# Sources and Types of Social Supports and Their Association with Mental Health Symptoms and Life Satisfaction among Young Adults with a History of Out-of-Home Care

**DOI:** 10.3390/children9040520

**Published:** 2022-04-06

**Authors:** Rhiannon Evans, Colleen C. Katz, Anthony Fulginiti, Heather Taussig

**Affiliations:** 1Centre for Development, Evaluation, Complexity and Implementation in Public Health Improvement (DECIPHer), School of Social Sciences, Cardiff University, Cardiff CF10 3BD, UK; 2Silberman School of Social Work, Hunter College, City University of New York, New York, NY 10065, USA; colleen.katz@hunter.cuny.edu; 3Graduate School of Social Work, University of Denver, Denver, CO 80208, USA; anthony.fulginiti@du.edu (A.F.); heather.taussig@du.edu (H.T.); 4Kempe Center, University of Colorado School of Medicine, Aurora, CO 80045, USA

**Keywords:** adolescent, young adult, foster care, social support, relationships, mental health, wellbeing, life satisfaction

## Abstract

Young adults with a history of out-of-home care report poorer mental health and life satisfaction compared to non-care-experienced peers. Social support is a known protective factor for mental health. There is limited evidence, however, on the relationship between sources (e.g., family members) and types (e.g., information) of social support and mental health symptoms and life satisfaction in this population. Reporting cross-sectional survey data from 215 young adults aged 18–22 years with a history of out-of-home care, the current study conducted descriptive, bivariate, and linear regression analysis to examine the different sources and types of support young adults receive and their relation to mental health symptoms and life satisfaction. Participants had high levels of support from family members, friends, and other adults. Most participants had informational support, but less than half had consistent material support. Regression analyses demonstrated that having enough informational and material support were associated with fewer mental health symptoms. Having family support and material support were associated with greater life satisfaction. Further longitudinal research is needed to understand the trajectory between social supports and mental health functioning and life satisfaction.

## 1. Introduction

The mental health and life satisfaction of young adults with a history of out-of-home care is a public health priority. Rates of mental illness are high for adolescents in foster care (>50%), with diagnoses of major depression and other mood disorders being the most prevalent [1]. Systematic reviews have reported that children in out-of-home care present with higher levels of psychopathology when compared to a community of matched samples [2]. A UK longitudinal study found that individuals had excess mortality in adulthood up to 42 years after reporting foster care and/or residential care status in the national census [3]. This increased risk was attributed to non-natural causes of self-harm, accidents, and other mental health and behavioral factors. Children and young people who have foster care experience have also reported lower rates of subjective wellbeing than those who have never been removed from their homes of origin [4].

Evidence exploring trajectories of symptoms of poor mental health and life satisfaction indicate that key risk factors often originate with early exposure to maltreatment [5,6,7], which is frequently the reported reason for care entry. The impact of the care experience, especially the type of care placement, in mitigating or increasing mental health problems is not yet fully understood [8]. Development of mental health problems in childhood and adolescence can continue into later life [9], although there is also evidence of stabilization in positive behavioral adjustment [10]. Mental health problems are also associated with a range of other adverse outcomes, notably lower levels of educational attainment and engagement [11,12].

Social support is a well-established protective factor for mental health and life satisfaction; adolescents and young adults with high levels of perceived social support tend to have lower levels of mental illness (specifically depression and anxiety) than their peers [13,14]. Social support can be understood as the perceived or received assistance that an individual has from other people [15]. It is a multi-dimensional construct that can encompass different types of assistance [16,17]. This can include informational, instructional, emotional, instrumental, and advocacy [18], although research often focuses on informational (e.g., advice), material (e.g., tangible), and emotional (e.g., esteem, affection and belonging) supports [19].

Young adults in out-of-home care tend to experience a paucity of all types of support [20,21]. Evidence from the California Youth Transitions to Adulthood Study (CalYOUTH) indicated that among 19-year-old individuals in foster care, 40% stated that they did not have enough people to turn to for emotional support, nearly half did not have enough people to provide material support, and more than 30% did not have enough people to give them advice and guidance [22].

Where available, support may be derived from a range of sources. In the general population, support from family members is often described as important for positive mental health [18,23]. Similar evidence has been reported for individuals with out-of-home care experience [24]. Equally, research with care leavers indicates that a lack of family support adversely affects life satisfaction [25]. Analysis of the National Survey on Child and Adolescent Well-Being II (NSCAW), a nationally representative longitudinal survey of children and families who have been investigated by Child Protective Services in the United States, found that having current contact with birth mothers and fathers was associated with fewer mental health symptoms [26]. However, minimal research has explored the types of support offered by family members. A recent systematic review recognizes that few studies have identified the types of support provided by birth parents, and the benefits that may emerge from the range of support offered [27].

Other family members may also be important to young adults with a history of care. Recent research has found that having positive relationships with foster parents is associated with higher levels of life satisfaction [28]. Despite more limited research evidence, there are also potentially important considerations regarding the impact of sibling relationships and supports. Sibling co-placement, and having the opportunity to sustain a relationship, is reported to be a protective factor for a range of mental health and wellbeing outcomes [29]. Meanwhile, sibling separation and a lack of a supportive relationship are considered to impede development and negatively impact mental health [29].

Although family members may be an important source of support, youth who have been in out-of-home care are less likely than their same-aged peers in the general population to receive support from their biological parents [30]. Qualitative research has suggested that children and adolescents in foster care can find it challenging to maintain positive relationships with their birth families [31,32,33]. As such, they often draw upon a wide range of other social supports, such as peers, teachers, and social care professionals [19,22,34,35,36,37]. These relationships are also found to be largely protective of mental health and life satisfaction. One qualitative study conducted with youth with histories of foster care involvement explored the role of natural adult mentors in supporting mental health during their transition to adulthood, recognizing the importance of consistent, mutual, and empathetic relationships that offer emotional, informational, and material support [19]. However, there has been limited consideration of how these other sources of support compare to family support in terms of being a protective factor for mental health and life satisfaction.

While there is an emerging evidence base on the association between different social supports, mental health, and life satisfaction for young adults with a history of out-of-home care, several gaps need to be addressed. First, there are few studies that report both the range of sources (e.g., family or other adult) and types (e.g., informational or material) of social supports. Second, the relative contribution of different sources and types of social support in protecting mental health and life satisfaction is not clear, and there is a need to further understand how different forms of assistance are associated with outcomes over and above other forms of social supports.

### Research Questions

The current study explores the sources and types of social support among young adults (ages 18–22 years) with a history of out-of-home care. The study further examines the extent to which these social supports are associated with mental health symptoms and life satisfaction.

Specifically, this exploratory study addresses the following research questions:Who and what are the sources and types of social support for young adults with a history of out-of-home care?What are the bivariate relationships between different sources/types of social supports and current mental health symptoms and life satisfaction?Are certain sources or types of social support associated with fewer mental health symptoms and life satisfaction over and above other sources/types of social support and relevant control variables?

## 2. Materials and Methods

The present study reports cross-sectional analysis of data from the longitudinal Fostering Healthy Futures (FHF) study, which was conducted in the United States.

### 2.1. Study Participants

The study includes data from eight cohorts of youth enrolled in the FHF intervention between 2002 and 2009. Participants were eligible for the study if they met the following inclusion criteria at baseline: (1) aged 9–11 years old; (2) had been placed in out-of-home care in the previous year by a participating county child welfare department; and (3) were living in out-of-home care at the time of the baseline interview.

For the current study, 243 young adults from the original FHF study who were between ages 18 to 22 years old were recruited to complete a long-term follow-up survey. The survey was completed an average of 9.4 years after the participants’ baseline survey. Of the 243 individuals recruited, 215 (88.5%) were located and consented to be interviewed. 7 participants declined the interview, 8 aged out of eligibility, and 13 could not be located or recruited.

About half (47.9%) of study participants identified as female. Participants’ mean age was 19.5 years old (SD = 0.94). For race and ethnicity, 54.0% self-identified as Latinx/Hispanic, 48.8% as White, 28.8% as American Indian, and 27.4% as Black. Participants had the option to identify more than one racial/ethnic category. More than a third (35.8%) of the participants were currently living in their own place; 18.1% were living with one or more biological parents; 15.8% were living in a relative’s home; 15.8% were living in the home of another adult (i.e., adoptive parent, family friend, and significant other’s parent); and the remainder were unhoused or living in a shelter, group home, treatment facility, college dorm, or prison.

### 2.2. Procedures

The protocol was approved by the Institutional Review Board and participants provided written consent for their participation. Most interviews took place in a face-to-face interview format (or by telephone when participants lived too far) and interview questions were read aloud by graduate student research assistants. Participants were compensated US $100 for completing an interview.

### 2.3. Measures

#### 2.3.1. Social Support (Independent Variables)

Questions from the Jim Casey Youth Opportunities Initiative survey were used to assess current sources and types of social support.

Sources of social support: Three questions were asked about the presence/absence of three groups of supportive persons (adult family members, good friends, and other adults) in their lives. Example: ‘Is there an adult in your family (not a spouse or significant other) that you will always be able to turn to for support (for example, to help you with a problem, to listen when you’re upset)?’ Each of the three questions had a binary response option of yes = 1 or no = 0. When participants responded “yes” to family support, follow-up questions included: Which one adult family member do you turn to most often?; How often do you see or communicate with this adult family member?; and How much can you count on this adult family member to provide you with the support you need? When participants responded “yes” to friend support, follow-up questions included: How many friends do you have that you can count on for support?; and How much can you count on these friends to provide you with the support you need? When participants responded “yes” to other adult support, follow-up questions included: Which one adult other than a family member do you turn to most often?; How often do you see or communicate with this person?; and How much can you count on this person to provide you with the support you need?

Types of social support: Two questions were used to assess the types of support available to participants: When you need someone to give you good advice about a crisis are there…?; and When you need someone to loan you money in an emergency, are there…? These two types of supports were classified as informational and material support, respectively. Each of the two questions had three response options of: enough people you can count on; too few people you can count on; and no one you can count on. The three response options were dichotomized into a composite variable of enough people you can count on = 1 vs. too few people or no one = 0.

#### 2.3.2. Mental Health Symptoms and Life Satisfaction (Dependent Variables)

Mental health symptoms: The K6 Scale was used to assess mental health symptoms. It is a six-item measure of serious mental illness and was developed with support from the U.S. Government’s National Center for Health Statistics for use in the redesigned U.S. National Health Interview Study [38]. The scale was designed to be sensitive to nonspecific distress to maximize the ability to discriminate cases of serious mental illness from non-cases. The K6 demonstrated high internal consistency and reliability across different demographic groups (Cronbach’s α in current study = 0.86). Each of the six items (e.g., During the past 30 days, about how often did you feel nervous? How often did you feel so depressed that nothing could cheer you up?) are rated by the respondent on a five-point scale from None of the time = 0 to All of the time = 4. A mean score was calculated, resulting in participant scores that ranged from 0–3.33 (M = 1.03, SD = 0.87). This measure was not administered to the first 22 participants in the study and therefore *n* = 193 in analyses using this variable.

Life satisfaction: Life satisfaction was assessed with one item from the project-modified Delighted-Terrible Scale [39]: ‘And last, a very general question, how do you feel about your life as a whole?’ The question had a 1–5 scale response option, with 1 = mostly unhappy and 5 = mostly happy (M = 4.33, SD = 0.98).

#### 2.3.3. Control Variables

Gender: Since source, type, and amount of support may differ by gender, this variable (operationalized as female = 0; and male = 1) was included in analyses [40].

Living history: Participants provided information regarding their living situation that included whether they had ever reunified with their birth parents (45.2%), lived with kin (87.9%), lived in non-relative foster care (75.8%), lived in a congregate care setting (52.6%), been adopted (27.2%), and/or emancipated from care (26.5%). All living history variables were independently coded (not experienced = 0 or experienced = 1), so participants could endorse multiple living experiences.

Mental health diagnosis: Participants were asked ‘Have you ever received a mental health diagnosis?’ The question had a binary response option of no = 0 or yes = 1. About a third (31.8%) reported having a mental health diagnosis at some point in their lives.

### 2.4. Analysis

The analytic strategy included three steps. First, descriptive analyses were conducted to summarize the characteristics of study participants and the source and types of social supports they received. Second, bivariate analyses (i.e., independent samples *t*-test) were performed to examine the unadjusted associations between independent variables (i.e., sources/types of social support), dependent variables (i.e., mental health symptoms and life satisfaction), and control variables. Finally, separate multiple linear regression analyses were conducted for mental health symptoms and life satisfaction. All independent and control variables that were significant in the bivariate analyses were simultaneously entered into the regression model to examine the adjusted associations between each independent variable and dependent variable while controlling for one another (i.e., a forward-selection model building approach).

Notably, although a data-driven criterion was used to build the final regression models, all variables in the current study were selected for inclusion based on a previous research or the researchers’ professional and lived experience with child welfare-involved populations. A sensitivity analysis was also performed to assess the robustness of our results to different data-driven model building strategies; this involved repeating our regression analyses using a backward selection model building approach, which produced the same pattern of statistically significant and non-significant associations (not included but available from the first author upon request). An alpha level of *p* < 0.05 was used to determine statistical significance, with *p* < 0.10 indicating a statistical trend; statistical trends were given consideration because of the limited research in this subject domain and our related concerns about Type 2 errors. *p*-values were used in a descriptive manner. Analysis was conducted in SPSS Version 25.

## 3. Results

### 3.1. Sources and Types of Social Supports

Descriptive statistics for participants’ sources of support, frequency of communication, and reliability of support are presented in Table 1. For family support, almost all of the participants reported having an adult family member that they could turn to for support. When asked which one family member they turn to most often, a third selected a birth parent and a third selected an extended family member. The remaining options, reported in order of frequency, were: adult siblings; adoptive family; foster family; legal guardian; and other. A quarter of participants lived with the named family member, a third communicated with the family member every day, and a fifth communicated with them two to five times per week. Almost two-thirds of participants reported that they could “always” count on this family member to provide the support needed, with an additional third of participants reporting that they could count on this family member most of the time.

For friendship-based support, three-quarters of participants said that they had good friends whom they could turn to for support. Almost three quarters had one to four friends, while a fifth selected having five to nine friends. Almost half of participants stated that they could always count on their friends, while two-fifths selected being able to count on them most of the time. For other adult support, over half of participants stated they had a non-family adult to turn to for support when they needed it. When asked which adult they relied on, a quarter reported being able to rely on a family friend/neighbor and a fifth selected a work colleague. The remaining options, reported in order of frequency, were: teacher/coach; non-relative mentor; adult from faith-based community; caseworker; staff person from residential facility or group home; lawyer; and other. A third of participants stated they communicated with this adult almost every day, a third had contact two to five times per week, and a third reported being in touch once a month to once a week. In total, half of participants said they could always count on this non-family adult, while a third said they could count on them most of the time.

Participants indicated the availability of informational support (e.g., someone to give good advice about a crisis) and material support (e.g., someone to loan money in an emergency) (Table 1). Almost three-quarters of participants stated they had enough people to count on for informational support. Less than half of participants selected having enough people for material support.

### 3.2. Bivariate Associations between Source and Type of Social Support, Mental Health Symptoms and Life Satisfaction

*T*-tests were used to explore the unadjusted associations between different sources and types of social support, mental health symptoms, and life satisfaction. Participants who had adult family members to turn for advice and support had fewer mental health symptoms (*t* = −3.9; *p* < 0.001) and higher life satisfaction (*t* = 2.9; *p* < 0.01). Having good friends was associated with fewer mental health symptoms (*t* = −2.5; *p* = 0.01), but not life satisfaction. Having another adult to turn to for advice and support was not related to mental health symptoms, but there was a trend towards it being related to higher life satisfaction (*t* = 1.7, *p* = 0.09).

Those who reported having enough people for informational support (namely those who provide good advice) had fewer mental health symptoms (*t* = −5.6, *p* < 0.001) and higher life satisfaction (*t* = 3.1, *p* = 0.002). Similarly, participants who reported having enough material support (i.e., people available to loan them money), had fewer mental health symptoms (*t* = −5.6, *p* < 0.001) and higher life satisfaction (*t* = 4.7, *p* < 0.001). In terms of the control variables, males had higher life satisfaction than females (*t* = 2.7, *p* = 0.007). Having a mental health diagnosis was related to more mental health symptoms (*t* = −3.1, *p* = 0.002) and lower quality of life (*t* = −3.6, *p* < 0.001). Experiencing non-relative foster care, adoption, congregate care, or living with kin were all unrelated to the dependent variables and were therefore not included as covariates in the multiple regression models.

### 3.3. Multiple Regression Models for Mental Health Symptoms and Life Satisfaction

Regression analyses were used to determine whether each source/type of social support was associated with mental health symptoms and life satisfaction over and above other support and control variables in the model (see Table 2). In the mental health symptoms model (*n* = 190), significant variables included: having enough people to give informational support (*b* = −0.40, *p* = 0.008); having enough people to give material support (*b* = −0.33, *p* = 0.013); and having a mental health diagnosis (*b =* 0.31, *p* = 0.014). Specifically, having enough informational and material support was associated with fewer mental health symptoms, whereas having a mental health diagnosis was associated with more mental health symptoms. There was a statistical trend (*p* = 0.08) for having a family member to turn to for support, which was associated with fewer mental health symptoms. The following variables were unrelated to mental health symptoms: having friends for support, having other adults for support, gender, and a history of emancipation or reunification.

In the life satisfaction model (*n* = 197), significant variables included: having a family member to turn to for support (*b* = 0.65, *p* = 0.004); having enough people for material support (*b* = 0.361, *p* = 0.015); having a mental health diagnosis (*b* = −0.37, *p* = 0.008); and gender (*b* = 0.32, *p* = 0.016). Specifically, having enough informational and material support was associated with higher life satisfaction, whereas having a mental health diagnosis was associated with lower life satisfaction. Males had higher life satisfaction than females. The following variables were unrelated to life satisfaction: having enough people to give informational support; having friends for support; having other adults for support; and a history of emancipation or reunification.

## 4. Discussion

### 4.1. Results

The present study examined the different sources and types of social supports available to young adults in out-of-home care. It further explored the extent to which these supports are associated with mental health symptoms and life satisfaction. Participants reported having a high number of social supports available to them, which included family members, good friends, and other adults. The majority of participants maintained that they could almost always or always count on family members and friends. This high prevalence of support availability runs counter to much of the existing evidence-base, which indicates that this population experiences a paucity of supports [20,21,22]. However, it does resonate with findings from other studies, which suggest that care-experienced young people derive support from a range of different sources [19,22,34,35,36,37].

A central finding from the study, and as reported in the wider evidence-base, is the protective role of family members for mental health and life satisfaction [24,25,26,41]. The availability of support from an adult family member was associated with fewer mental health symptoms and higher life satisfaction. Moreover, this support was related to life satisfaction over and above other types and sources of support. Notably, when identifying the specific family member, they are most likely to turn to for support, a third of participants cited a birth parent. Given the complexity of relationships that individuals in care can have with their biological parents [31,32,33], and the fact that they are less likely to receive parental support than peers in the general population [30], it is important to recognize the potential need for biological families to be integrated into young adults’ supportive social networks. However, there are risks of integration that need to be carefully attended to, such as the potential for trauma rearousal [42,43].

Non-family relationships were also considered vital to mental health and life satisfaction. Having good friends for support was associated with fewer mental health symptoms. Having a non-family adult support person was non-significantly associated with greater life satisfaction, although this association did not hold for mental health. Key adult support figures cited by participants included family/friends and neighbors, work colleagues, and teachers.

Despite the indication that participants had sources of support that could be counted on, these sources were not necessarily dependable for all types of support. While almost three quarters of participants maintained that they had access to informational support, less than half felt they had enough people to count on for material support. This finding aligns with results from the California Youth Transitions to Adulthood Study (CalYOUTH), which indicated that informational support was the most readily available type of support, but that material support was less frequent [22]. The present study similarly reports a lack of material support, with almost half of young adults not having enough people to offer this type of assistance. Both types of support were significantly associated with mental health symptoms and life satisfaction over and above other variables. This indicates a potential issue around young adults with out-of-home care experience not having access to the full range of supports that are required for positive mental health and life satisfaction.

### 4.2. Strengths and Limitations

The study has a number of key strengths. First, most prior research in this area has examined the role of social support among care-leavers who have emancipated from foster care. The current study’s sample consists of young adults with a range of living histories and current living situations. Second, the study addresses a key evidence gap; it considers both sources and types of social supports, which most research to date does not address simultaneously. Third, through the regression models, the study was able to control for potential confounding variables (e.g., gender, mental health diagnosis, and living history), in the attempt to isolate the association of social supports and the outcomes of interest.

There are also a number of limitations that should be considered when interpreting the results. First, this is a cross-sectional study. As such, the temporal ordering of social supports, mental health symptoms, and life satisfaction is not known. Therefore, it is not possible to infer if the sources and types of social support are a cause of any observed relationships with mental health and life satisfaction. Second, while the study benefitted from a high response rate, the sample size prevented further consideration of sources and types of support by gender, race, ethnicity, living situation history, and other sub-groups. It was also not possible to explore how support types differed by support source, and how this was associated with mental health symptoms and life satisfaction. Third, while the study considered both informational and material support, it did not consider additional types of support explored in the extant evidence-base, such as emotional support. Fourth, the study was reliant on self-report measurement, which may be subject to recall and reporting bias. Fifth, the measurement of several key constructs was based on a single item.

### 4.3. Future Directions

The findings from this study provide a number of useful directions for research, policy, and practice. In terms of research, the field would benefit from additional longitudinal studies devoted to the health and wellbeing of adolescents and young adults with child welfare experience. To date there is a wealth of longitudinal datasets that explore risk and protective factors for mental health and life satisfaction in this population around the world [44,45,46,47,48]. However, for the large part they have not reported analysis of the relationships between/among sources and types of social supports and mental health status. Such analyses could shed light on the importance of certain types of support in the lives of young adults, particularly during the transition from care to independent adulthood. Further, there have been some strong early qualitative studies investigating social support for youth transitioning from care [49]. It would be beneficial to conduct additional qualitative research with this population to specifically investigate when different sources and types of social supports are most useful across different developmental stages, and how these supports may be related to mental health symptomatology and management.

At the policy level, there is a range of legislation and directives internationally that can continue to foreground and prioritize high quality social supports. In the United States, the Family First Prevention Services Act (FFPSA) was enacted in 2018 to keep youth with their families/communities of origin and out of the foster care system by increasing access to community psychiatric health and substance abuse services. Meanwhile, in the UK, the Children and Social Work Act stipulates a relationship-based approach to social work that fosters positive relationships, particularly between social workers and children [50]. Research evaluating the extent to which these policies have impacted perceived social support in child welfare-involved youth would be valuable.

In terms of future social care practice in relation to young people, it is important to provide opportunities to develop and sustain positive social relationships. In the USA, the Fostering Connections to Success and Increasing Adoptions Act (Fostering Connections) passed in 2008 includes a state requirement relating to social support: child welfare administrators must identify “relatives” (either biological or social) who can serve as supports when youth are removed from their homes. Equally, guidance by the UK National Institute for Health and Care Excellence (NICE) recommends that organizations, practitioners, and foster carers work to ensure that children and young people in care and leaving care have nurturing relationships in order to reach their potential [51].

There are a number of interventions in this area, including those that support young people’s fostering of positive connections with a range of individuals [52,53,54]. Furthermore, there is a need for research, policy, and practice to understand how to best provide continuity in social networks, potentially through support for placement stability and reunification so that young adults can remain connected to their communities of origin [51].

## Figures and Tables

**Table 1 children-09-00520-t001:** Description of Source and Types of Social Supports (*n* = 215).

	Number (*n*)	Percentage (%)
Family Support		
Availability of Familial Adults		
Yes	189/214	88.3
Birth parent	65/188	34.6
Extended family member	65/188	34.6
Adult sibling	27/188	14.4
Adoptive family member	21/188	11.2
Foster family member	5/188	2.7
Legal guardian	2/188	1.1
Other	3/188	1.6
Frequency of Communication with Familial Support		
Lives with family member	51/188	27.1
Almost everyday	62/188	33.0
Less than 2–5 times per week	41/188	21.8
Once per month to once per week	30/188	16.0
Once per year to every few months	3/188	1.6
Less than once per year	1/188	0.5
Reliability of Support		
Always	113/188	60.1
Most of the time	55/188	29.3
Sometimes	18/188	9.6
Not very often	2/188	1.1
Friend Support		
Availability of Friends		
Yes	163/215	75.8
Number of good friends		
1–4 friends	117/163	71.8
5–9 friends	34/163	20.8
10+ friends	12/163	7.3
Reliability of Support		
Always	75/163	46.0
Most of the time	68/163	41.7
Sometimes	17/163	10.4
Not very often	3/163	1.8
Other Adult Support		
Availability of Other Adults		
Yes	117/215	54.4
Source of Other Adult Support		
Family friend/neighbor	31/117	26.5
Work colleague	25/117	21.4
Teacher/coach	17/117	14.5
Non-relative mentor	8/117	6.8
Adult from faith-based community	5/117	4.3
Caseworker	3/117	2.6
Staff from residential home	3/117	2.6
Lawyer	1/117	0.9
Other	24/117	20.5
Frequency of Communication with Adult Support		
Almost everyday	35/117	29.9
Less than 2–5 times per week	25/117	21.4
Once per month to once per week	39/117	33.3
Once per year to every few months	15/117	12.8
Less than once per year	3/117	2.6
Reliability of Support		
Always	60/117	51.3
Most of the time	40/117	34.2
Sometimes	16/117	13.7
Not very often	1/117	0.9
Types of Support		
Informational (i.e., Advice)		
Enough people	156/215	72.6
Too few people/No one	59/215	27.4
Material (i.e., Money)		
Enough people	103/215	47.9
Too few people/No one	112/215	52.1

**Table 2 children-09-00520-t002:** Linear Multiple Regression Analysis of Mental Health Symptoms and Life Satisfaction.

	Mental Health Symptoms			Life Satisfaction		
	*b*	*SE*	*p*-Value	*b*	*SE*	*p*-Value
Source of Support						
Familial Adults	−0.366 ^†^	0.205	0.076	0.647 **	0.220	0.004
Friends	−0.147	0.139	0.292	0.010	0.153	0.949
Other Adults	0.074	0.116	0.521	0.138	0.127	0.276
Type of Support						
Informational (i.e., Advice)	−0.399 **	0.148	0.008	0.013	0.163	0.935
Material (i.e., Money)	−0.334 *	0.133	0.013	0.361 *	0.147	0.015
Control Variables						
Gender	−0.119	0.118	0.316	0.315 *	0.130	0.016
Emancipation	0.046	0.141	0.745	0.144	0.155	0.353
Reunified	0.119	0.120	0.326	−0.016	0.133	0.903
Mental Health Diagnosis	0.308 *	0.124	0.014	−0.365 **	0.136	0.008
Model Fit						
Adjusted R^2^ Value	0.211			0.175		

^†^*p* < 0.10; * *p* < 0.05; ** *p* < 0.01.

## Data Availability

Some of the data presented in this manuscript have been archived at ICPSR: https://www.icpsr.umich.edu/web/NACJD/studies/36880, accessed on 4 Februray 2022.

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
