# Peer review of "Sources and Types of Social Supports and Their Association with Mental Health Symptoms and Life Satisfaction among Young Adults with a History of Out-of-Home Care"

_children, 2022, doi:10.3390/children9040520_

Round 1
Reviewer 1 Report
I would like to thank the Editor for inviting me to review this manuscript. It is a generally well-written manuscript reporting the results of a cross-sectional study. The authors found associations between mental health symptoms and life satisfaction and social support (different types and sources).
Major comment
My major comment is about the variable selection procedure. Briefly, you conducted bivariate correlational analyses and the variables with significant correlation were included in the multiple regression model. This variable selection procedure is data-driven rather than theoretically based. Perhaps the theoretical basis is lacking at this point and that basis needs to be established. In this case, I prefer a larger model with all possible control variables. Fewer effects are overlooked with this approach. If the number of independent variables is too large in relation to the number of participants, or multicollinearity occurs, backward selection is possible, which has further disadvantages, but fewer disadvantages than your procedure. Please show the results of the regression model with all possible control variables. If there is a sufficient theoretical basis, please justify your choice of variables theoretically and then build your model
Minor comments
Materials and Methods section
Line 111: In which country was the Fostering Healthy Futures study conducted? In the appendix, the authors state that “this research study is funded by the National Institute of Justice, the Kempe Foundation, and other local foundations.” Please include this information in the manuscript and add a statement about the role of the funder for this study.
Line 135: What does IRB mean?
Analysis section
Line 196: Which software package do you use for your analyses?
Line 206: Please add that p values are used in descriptive manner. This is because the study is exploratory and no adjustment for multiplicity is made.
Results section
Line 246, table 1:
Please indicate the population for each category of each variable. A possible form could be:
Availability of Familial Adults Yes 189/214 88.3
For yes-no questions, I think it is sufficient to show only the yes answers or only the no answers.
Description of both outcomes (life satisfaction and mental health problems) is missing. Please add a description for the outcomes (e.g. mean and standard deviation).
Line 300, table 2:
How many participants are used in each model?
The F value with three stars is not appropriate to consider the model fit. Please delete this line.
The reporting of the r-squared is not useful here, since hypotheses about associations between variables are to be tested and not models compared with each other. Is the r-squared the corrected r-squared? Please use the corrected r-squared or indicate that it is the corrected r-square. I also agree if you delete the line.
Line 280: I think, the results under the caption of mental health symptoms are the results of life satisfaction and vice versa. Please add an assessment of the relevance of the associations found.
Discussion section
Line 340/341: Please use the word association instead of predictor.
Reviewer 2 Report
The manuscript is well-written in general, and the authors did a good job of laying out the association between social support, mental health, and life satisfaction in a group of young adults who had previously been in out-of-home care. Nevertheless, I think the article could be improved if the comments below were addressed:
Comment 1: Young adulthood is a period of development during which significant changes take place. As such, authors could improve the manuscript from a developmental perspective by reviewing and citing relevant studies on the developmental trajectories of mental health and life satisfaction from adolescence into young adulthood as well as their associations with outcomes in young adulthood.
Comment 2: Why did the authors select to assess life satisfaction in young adults using one item from the project-modified Delighted-Terrible Scale when there are so many other instruments available? (For example, SWLS; Diener et al., 1985).
Comment 3: The implications for research and practice must be discussed.
Comment 4: I'm concerned about the use of categorical variables (such as gender) in bivariate associations with continuous variables. A table can also be used to present bivariate associations.
It was a pleasure to review this manuscript and I wish the authors the very best with their revisions.
